# Biodiversity of Root Endophytic Fungi from *Oxyria sinensis* Grown in Metal-Polluted and Unpolluted Soils in Yunnan Province, Southwestern China

**DOI:** 10.3390/plants10122731

**Published:** 2021-12-11

**Authors:** Meiyan Zhu, Yanhua Ding, Xuejiao Li, Yuqing Xiao, Zhiwei Zhao, Tao Li

**Affiliations:** 1State Key Laboratory for Conservation and Utilization of Bio-Resources in Yunnan, Yunnan University, Kunming 650091, China; zhumeiyan93@126.com (M.Z.); dingyanhua09@163.com (Y.D.); lixuejiao0903@163.com (X.L.); xiaoyuqing@mail.ynu.edu.cn (Y.X.); 2School of Life Sciences, Yunnan University, Kunming 650500, China

**Keywords:** *Oxyria sinensis*, dark septate endophyte (DSE), arbuscular mycorrhizal fungi (AMF), diversity, metal contamination

## Abstract

*Oxyria sinensis* adopts a tolerant strategy as a metal excluder to survive toxic metal concentrations. Biodiversity and the endophytic fungal community colonizing the *O. sinensis* roots were assessed from a mining area (MA) and a neighboring non-mining area (nMA) in southwestern China. All *O. sinensis* roots formed fully developed dark septate endophytes (DSEs) and arbuscular mycorrhizal fungi (AMF). Total DSE colonization was higher for the MA versus nMA, in contrast to the total AMF colonization in the two sites. The DSE colonization was higher than AMF colonization regardless of the site. Pure-culture data showed that the fungi closely related to *Exophiala*, *Cadophora* and *Phialophora* dominantly colonized the *O. sinensis* roots. A total of 450 operational taxonomic units (OTUs) were identified showing the presence of a distinct fungal community in MA and nMA, which was shaped by soil physiochemical properties, including soil Zn concentrations and organic matter. We found that *O. sinensis* accumulates and adapts efficiently to local endophytic fungi to achieve the expansion of its community, including the spontaneously reclaimed DSE. This property may be targeted to achieve its colonization with a pioneer plant for phytoremediation in the restoration of a vegetation cover in a metal-contaminated area.

## 1. Introduction

Soil metal pollution is a global concern, due to its high toxicity to various living organisms. This is especially true for barren mining/smelting areas with excessive leaching of heavy metals [1,2]. A successive vegetation cover can provide the necessary surface stability to reduce the dispersion of contaminants via eolian spread and water erosion. Meanwhile, vegetative technologies such as phytoremediation are increasingly recognized as cost-effective and ecologically sound alternatives to metal-polluted land rehabilitation [3,4].

Heavy-metal-contaminated soils suffer from macronutrient and substrate deficiencies, limiting the option of revegetation, which is an otherwise ideal approach for remediation [5,6]. A growing body of literature from both field and greenhouse experiments provides evidence that root-associated endophytes, e.g., the most commonly encountered dark septate endophytes (DSEs) and arbuscular mycorrhizal fungi (AMF), do indeed aid some specific plants in adapting and resisting a wide range of biotic and abiotic environmental stresses [6,7]. For example, Hui et al. report that the colonization of a root-associated endophytic fungus of Sebacinales, *Serendipita* (=*Piriformospora*) *indica*, confers cadmium (Cd) tolerance to *Nicotiana tabacum* in hydroponics, pot and field trials [8]. Previous data from greenhouse experiments show that the colonization of maize roots with DSE was associated with an increased Cd tolerance, significantly less phytotoxicity and enhanced plant growth [9]. Subsequent field trials showed evidence of an increased plant survival rate in mine spoils due to root colonization with mycorrhizal fungi, e.g., three *Acacia* species, *Pinus sylvestris* [10,11]. Additionally, diverse and multifunctional root-associated endophytes are believed to be major determinants of plant community structure in early succession [12,13] and enhance plant establishment in contaminated soils [10,14]. Long-term studies were carried out by Veselkin et al. [15], spanning 1978–2006 to evaluate the succession of vegetation on different parts of overburdened rocks in Karaganda, Kazakhstan. The authors noted that the association of obligate species with AMF during the revegetation phase led to enhanced species diversity. The mycorrhizal plants’ accelerated revegetation and reclamation were suggested to involve different strategies, such as amelioration and adaptation, which subsequently encouraged the establishment and vegetation development. In addition, the synergistic interactions between plants and the root-associated endophytes led to better nutrient and water access by stimulating initial growth and improving plant biomass production, thereby enhancing the rate of remediation of contaminated soil [16]. Overall, root fungal bioaugmentation, in combination with plant reclamation represents an attractive approach for the decontamination of soils polluted with heavy metals [6].

There are many ancient and abandoned slag heaps of non-ferrous metals from mining in Yunnan province, southwestern China, and successive vegetation covers have been established spontaneously in these slags at different times. These indigenous plants are potential reservoirs of germplasm for phytoremediation purposes [17,18], as these are not only tolerant to the stressed environment but are also adapted to the prevailing local climate conditions. Among these, an herb species native to the Himalayas known as *Oxyria sinensis* (Polygonaceae) has attracted a lot of attention for potential revegetation, due to its high metal tolerance and the ability to generate considerable biomass, as well as the more easily accessible sexual seeds and asexual rhizomes [19,20]. Interestingly, *O. sinensis* adopts a tolerant strategy as a metal excluder to survive in toxic metal concentrations presented in the growth medium [21]. However, there is little knowledge about the interactions between *O. sinensis* and their root-associated endophytic fungi, which are actively effective for the integral role in response to heavy metals. Our work aims to determine the colonization characteristics and community structure of root-associated endophytic fungi from both mining and non-mining ecotypes of *O. sinensis*, using pure-culture and culture-independent methods. In addition, we aimed to explore the potential functional roles of endophytic fungi in enhancing their host plant tolerance/adaptation against metal stress and the potential restoration abilities of *O. sinensis* in the Yunnan unclaimed mining area, China.

## 2. Results

### 2.1. Chemical Properties of the Soil

The physiochemical properties of the soil from the Kuangshan mining area (MA) differed from those of the non-mining area (nMA). The soil from the mining area had significantly higher concentrations of Pb, Zn and Cd compared to the non-mining area and contained 191.9 ± 37.67 mg kg^−1^ Pb, 719.5 ± 210.00 mg kg^−1^ Zn and 1.9 ± 0.51 mg kg^−1^ Cd, respectively. Similarly, there was a significantly higher organic matter content in MA, but a markedly lower content of available P (19.7 ± 0.84 mg kg^−1^ in MA vs. 26.1 ± 1.06 mg kg^−1^ in nMA) (*p* < 0.05). There was no significant difference in the alkaline hydrolysis N or the available K and soil pH for both areas (Table 1).

### 2.2. DSE and AMF Colonization

Root samples from both areas (MA and nMA) showed the presence of typical AMF structures, i.e., arbuscules, hyphal coils and vesicles, as well as DSE structures such as dark septate hyphae and microsclerotia (Figure 1). For both areas, we found that there was a higher DSE colonization than that of AMF in the roots of *O. sinensis*, and this was especially true for MA samples with a significantly higher DSE colonization (*p* < 0.01). Furthermore, we also observed that metal pollution of MA soil significantly inhibited the AMF colonization, and AMF colonization intensity in the nMA was markedly higher than its mining counterpart (*p* < 0.05), while DSE colonization exhibited a reverse trend (Figure 2). AMF and DSE colonization were 2.90 ± 0.55% and 11.46 ± 1.87% in the MA area and 6.28 ± 1.00% and 8.97 ± 1.39% in the nMA area, respectively.

### 2.3. DSE Isolation and Identification

A total of 48 root-associated melanized fungi were isolated from the roots of *O. sinensis*, including 29 and 19 strains isolated from the non-mining and mining ecotypes, respectively. According to phylogenetic analysis, these fungal strains were clustered into fourteen phylogenetic groups, including five MA-specific clades, four nMA-specific clades and five clades overlapped in both areas (Figure 3). We also found that the majority of the isolates were closely related to fungal members of *Cadophora*, *Phialophora*, *Leptodontidium* and *Exophiala*, and the most dominant fungal strains belonged to the clades of Helotiales, Chaetothyriales, Ascomycota and *Exophiala*, accounting for an approximate total relative abundance (RA) of 60% (Figure 3). The relative abundance of other several clades, including *Minimelacolocus*, *Pyrenochaeta, Acrocalymma*, Sordariomycetes 2 and Pleosporales were also above 4%, but the species related to Magnaporthaceae, *Tetraploshaeria* and Lophiostomataceae had the lowest proportions.

### 2.4. Endophytic Fungi Diversity and Community Structure in O. sinensis Roots

In this study, 345 and 313 fungal operational taxonomic units (OTUs) colonizing the roots of *O. sinensis* from the MA and nMA were found, respectively. There were 208 shared OTUs, which accounted for a proportion of 60.29% and 66.45% in MA and nMA, respectively (Figure 4). The OTUs’ abundance change curve flattened when the sequencing depth was increased, showing that the data covered the majority of the bioinformatic information about root endophytic fungi (Figure 5). The ‘richness’ (Chao1 and ACE) and ‘evenness’ (Shannon and Simpson) of the dominant broad fungal groups were calculated (Table 2). *O. sinensis* in MA had a greater tendency for richness, while no significant differences in evenness were observed between MA and nMA.

The endophytic fungi community structure (genus level) of *O. sinensis* from MA was richer than in nMA. *Xylaria* displayed the highest relative abundance at genus level, accounting for 63.5% and 79.8% in MA and nMA, respectively. This was followed by the fungi of *Magnaportiopsis*, Pleeosporales, Ascomycota, *Sparassis* and *Llyonectria* in MA. However, these fungal members had less than 0.5% relative abundance in nMA (Figure 6). Subsequently, the fungal community structures in the *O. sinensis* roots from MA and nMA were examined using Principal Coordinate Analysis (PCoA). It was performed based on the unweighted Unifrac distance matrixes, and the two axes maximally reflected 47.50% of the total difference (PC1, 29.17%; PC2, 18.33%) (Figure 7). Interestingly, the fungal communities in the roots of *O. sinensis* in MA were distinct from their nMA counterparts. Spearman’s correlation analysis showed that environmental factors highly correlated with species abundance (Figure 8); e.g., available ‘OM’ (organic matter), and Zn had a close correlation with seven and eight fungal genera, respectively, followed by Cd (six genera), available K (five genera), available P (four genera) and Pb (four genera) and available N (one genus). The number of environmental factors corelating with fungal abundance was in the following order. Fungi belonging to un--s-Trechisporales sp. correlated with six environment factors, followed by Un--s-Ceratobasidiaceae (five factors), Un--s-Chaetothyriales (four factors) and three environment factors for the other four fungal taxa (e.g., *Tetracladium*).

## 3. Discussion

*O. sinensis* (Polygonaceae) usually grows as a pioneer plant in artificially damaged environments, such as those affected by road construction, damming and mining sites. We observed that both ecotypes of *O. sinensis* growing naturally can form AM structures (hyphae, vesicles and arbuscules) in the roots, and simultaneously lead to the appearance of DSE colonization in large numbers in both MA and nMA. These findings suggest that AMF and DSE are countermeasures for the successful adaptation of the *O. sinensis* to such environments, with the colonization intensity of DSE being higher than AMF, regardless of the site. Furthermore, the DSE colonization was higher in the MA, while the AMF colonization presented the opposite picture. Ruotsalainen et al. [22] reported similar results in *Deschampsia flexuosa*, where the colonization intensity of AMF decreased with heavy metal stress, while the DSE colonization showed a slight increase, which was not found to be significant. The colonization of AMF in MA is usually lower than that of in the nMA and may also be related to the tolerance of AMF to heavy metals, as AMF is usually more sensitive to metals than DSE [23]. Gucwa-Prezpióra et al. [24] found that AMF colonization in *Plantago major* had a negative correlation with the bioavailable contents of heavy metal in the soil, while DSE colonization in *Salix caprea* was positively correlated with Pb concentration [25,26]. Besides this, it has been suggested that fungal endophytes are not merely mycorrhizal copycats, and DSE has more ecological functions in a stressful environment [6]. Similarly, Vergara et al. [27] reported that DSE contributes to tomato growth by increasing nutrient recovery efficiency, nutrient accumulation and the utilization of organic matter. *Leptodontidium orchidicola*, a species of DSE, has been claimed to improve tomato quality with an inorganic N source [28], as well as the enhancing P absorption by the maize seedlings [29]. Higher DSE colonization intensity in the mining area suggests that DSE may either be an integrated partner for *O. sinensis* to conquer the metal stress or *O. sinensis* is providing a shelter for the DSE in this stressful environment. AMF colonization intensity in the nMA was higher than in the MA, which may correlate with the addition of potassium, which significantly promoted the *Lablab purpureus* biomass, subsequently reduced heavy metal content and increased the AMF colonization intensity [30].

In this study, we isolated 48 DSE strains from the MA and nMA, and the fungi from the clades of Helotiales, Chaetothyriales and *Exophiala* were found to be most abundant. Many taxa in Helotiales and Chaetothyriales have been repeatedly noted as DSE fungi [31,32,33]. For example, our previous research also demonstrated that *Exophiala* was the dominant DSE fungi in the mining area [34]. Some *Exophiala* strains, such as *E. pisciphila*, have an inherently strong tolerance to heavy metals [35] and may also confer metal tolerance to the host plant upon root colonization [7,36,37]. We also noted that amongst the 14 fungal clades, only 5 clades consisting of 30 fungal strains (accounting for 62.5%) were overlapped in both areas, suggesting the environmental variations of soils contributed to the reshaping of root-associated endophytes. These results were also consistent with the previous findings that the physicochemical factors of metalliferous habitats, as the main physicochemical filters, strongly determine distinctive microbial communities of abandoned tailing dumps [38]. The results obtained by the culture-independent methods showed that *Xylaria* fungi are one of the most dominant root colonizers, which were reported to initiate endophytic associations with wide host ranges at a global level [39]. Subsequently, Wężowicz et al. [40] found that *Xylaria* sp. isolated from the roots of *Verbascum lychnitis* could significantly improve the photosynthesis rates of the plant-endophyte consortia on the post-mining waste dump containing elevated quantities of toxic metals (Pb, Zn and Cd).

Presumably, the need to combat severe and complex environmental stresses in the MA necessitates the co-existence of these pioneer plant species with endophytic fungi that can withstand heavy metal stress [10,11]. Since the energy and material sources of endophytic fungi were provided by the host plant, it follows that the amount of resources obtained by the host plant directly affects the amount of resources obtained by endophytic fungi. Previously, a study reported that the rate of colonization of a non-mycorrhizal plant, such as Cyperaceae with AMF, positively correlated with the soil N content and pH, while the contents of available P, K and organic matter had a negative correlation [41]. In our study, the endophytic fungi of *O. sinensis* in nMA were monotonous, compared to those in the MA. This may suggest that a high level of endophytic fungi species diversity can ‘guarantee’ plant productivity under different environmental conditions. Other studies also highlighted the positive effect of the species diversity of endophytic fungi on plant productivity and plant coexistence [42,43]. It has been reported that the complementary and selective effects of endophytic fungi lead to increased plant productivity, and that single-inoculation AMF can more effectively maintain the stable growth of plants.

## 4. Materials and Methods

### 4.1. Sample Sites and Collection

Root samples were obtained from an abandoned Pb and Zn slag heap (where smelting has been carried out for centuries using traditional methods [44]) with an area of 4 km^2^ in Huize country, located at 103°42′44″ E, 26°38′58″ N, Yunnan province, southwestern China. For hundreds of years, vegetation was spontaneously established on the spoil heaps, consisting of several pioneer plant species such as *Carex inanis* and *Cyperaceae hebecarpus*. Of the perennial herbaceous species, *O. sinensis* dominates the restoration in extremely metal-stressed sites [19].

In these sample sites, both ecotypes of *O. sinensis* from the mining area in Kuangshan and in the neighboring non-mining area (located at a distance of 27.6 km away) were studied. From each sample site, two sample plot replicates measuring 40 × 40 m^2^ were selected, and in total, 20 roots and soil sample repetitions were randomly collected for each *O. sinensis* ecotype during September 2018.

### 4.2. Soil Analysis

Soil samples (5–25 cm depth) were collected and allowed to air dry at room temperature (RT) for 2 weeks and kept for up to 2 months at RT until further treatment. Samples were milled into a fine powder, and a 0.25 mm nylon mesh was used to sieve out the larger pieces. Flame atomic absorption spectrometry (TAS-990 spectrophotometer, Beijing Puxi Instrument Factory, Beijing, P.R. China) was used to estimate the levels of Cd, Zn and Pb in soil samples as described previously [36]. As per the literature, the alkali N-proliferation method was used to calculate the alkaline hydrolyzed nitrogen [45], and the potassium dichromate oxidation-outer heating method was used to determine the soil organic matter [46]. Available phosphorus and potassium were calculated by the Olsen method and the ammonium acetate extraction, respectively [47]. The soil pH was measured using the electric potential method [46].

### 4.3. DSE and AMF Colonization

To assess the DSE and AMF colonization, the roots were adequately washed with running water and subjected to 10% (*w*/*v*) KOH treatment for 1 h in a 90 °C water bath. The samples were then stained with 0.5% blue ink (Hero^®^203, Shanghai, China) as described by Jing et al. [48]. At least 8 root fragments, 2 cm of each, were used to assess the intensity of DSE and AMF colonization, i.e., DSE hyphae and microsclerotia, and arbuscules and vesicles of AMF, under a compound-light microscope (OLYMPUS-BX31) [49,50]. If the presence of a microsclerotium or a dark septate hypha occurred in the microscopic field, the root sample was recorded as DSE colonization (AMF colonization was considered if an arbuscule or a vesicle was encountered.).

### 4.4. DSE Isolation and Identification

Thoroughly cleaned, fresh root samples were subjected to surface sterilization for 5 min each with 75% ethanol and 10% NaClO. The roots were then rinsed thrice with sterile distilled water, followed by air drying under sterile conditions [51]. These samples were then fragmented into 5 mm segments and deposited on both 1.8% potato dextrose agar (PDA, 200 g of potato extract and 20 g of dextrose per liter of water) and 1.8% malt extract agar (200 g malt meal extract per liter of water, pH 6.4), supplemented with 50 mg L^−1^ of ampicillin. The samples were incubated in the dark for 30 days at 28 °C. Root samples were monitored on alternate days for the growth of new melanized fungal colonies, which were then transferred to fresh PDA slants.

The melanized fungal isolates were identified using both morphological and phylogenetic methods, as previously described [52]. Microscopic identification was carried out for the sporulated isolates, which confirmed the presence of conidia or conidiogenous cells [53]. Using the internal transcribed spacer, ITS1-5.8S-ITS2 rDNA, sequences of all fungal isolates were cloned using a primer set ITS1-F/ITS4 [54]. Phylogenetic analyses of all fungal isolates and similar ITS sequences retrieved from GenBank were carried out to identify their taxa [55].

### 4.5. Root-Associated Fungal Assemblage Using Culture-Independent Methods

Root-associated fungal assemblage was assessed using culture-independent methods based on the next-generation sequencing of the fungal ITS1. Briefly, four replicate samples of the two *O. sinensis* ecotypes with the maximum DSE colonization intensity were selected. The fresh roots were thoroughly cleaned and surface-sterilized as described above, then subjected to total DNA extraction using PowerPlant Pro DNA Isolation Kit (MOBIO Laboratories, Carlsbad, CA, USA) immediately or after refrigeration storage (−86 °C). The fungal universal primer ITS5-1737F/ITS2-2043R was used to amplify the ITS1, fusion with the barcodes, respectively [55].

PCR was carried out in triplicate for each sample (10 ng of DNA templates) in a 30 µL reaction volume, with 15 µL of Phusion High-Fidelity PCR Master Mix (New England Biolabs, Ipswich, MA, USA) and 0.2 µM of e ach primer. The reactions were initiated at 98 °C for 1 min. This was followed by 30 cycles of denaturation at 98 °C with 10 s per cycle, annealing at 50 °C for 30 s and elongation at 72 °C for 60 s. The final elongation step was carried out at 72 °C for 5 min. Template DNA was excluded for the negative controls. PCR products were purified and separated by 2% agarose gel electrophoresis using a Qiagen gel-extraction kit (Thermo Scientific, Waltham, MA, USA). A TruSeq^®^ DNA PCR-Free Sample Preparation Kit (New England Biolabs) was used to construct genomic DNA libraries, which were then sequenced by Illumina HiSeq 2000.

The clean data were subjected to OTUs analysis at 97% similarity. Species–effort curves were made to determine the sensitivity of the observed richness vs. the number of sequencing. For ascertaining the diversity and richness of root-associated fungal communities in the two *O. sinensis* ecotypes, α diversity indices such as Chao1, Shannon, Simpson and ACE were carried out in QIIME (Version 1.7.0). Additionally, β diversity, including the un-weighted and weighted UniFrac distances, was studied using QIIME [55]. Sequencing results are available from the sequence read archive database, under GenBank accession number PRJNA593337.

### 4.6. Statistical Analysis

A *t*-test was carried out to analyze the difference among the two *O. sinensis* ecotypes, and differences were considered significant at *p* < 0.05. The correlations between environmental variables and abundant classes were analyzed by Pearson’s test using SPSS Software, version 19.0. PCoA was carried out using the weighted UniFrac distance. The α and β diversity indices between the two batches were assessed using a paired *t*-test and Wilcoxon test using the stats R package (Version 2.15.3). All parameters in the dataset of AMF and DSE colonization intensities were screened to detect outliers and were excluded from the dataset in each comparison [56].

## 5. Conclusions

*O. sinensis* growing naturally forms large number of AMF structures and DSE colonization in MA and nMA. The endophytic fungal community structures of *O. sinensis* in nMA were more monotonous than those in the MA. The present study suggests that to accumulate more local endophytic fungi, such as *Exophiala*, presumably contribute to the reclamation of *O. sinensis* in extremely metal-polluted soils, e.g., metal mine tailings, as well as the expansion of its own community. However, further study on the potential biological effects of the detected fungi, as well as other functional bacteria or other environmental conditions (e.g., essential elements) in the survival and tolerance of *O. sinensis* in mine-tailing soil is required, and there is also a need for more information on the potential for phytoremediation, which may be used in the restoration of vegetation cover in heavy-metal-contaminated areas.

## Figures and Tables

**Figure 1 plants-10-02731-f001:**
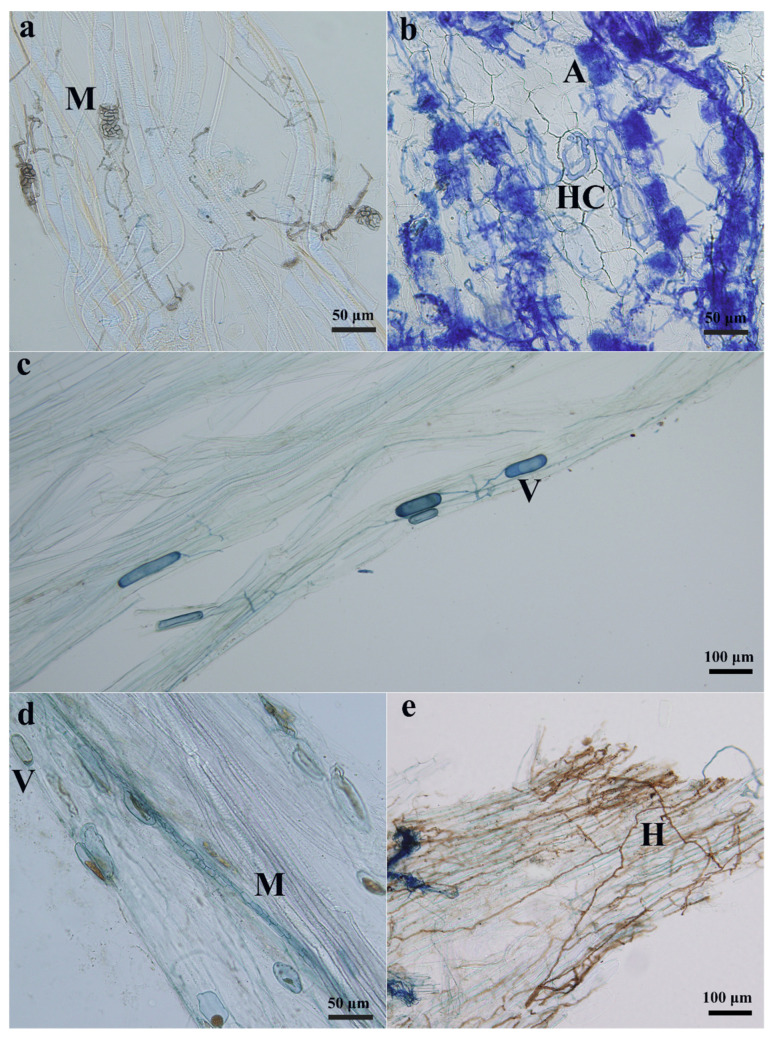
Microscopic examination of morphological characteristics of DSE and AMF root colonizers of *O. sinensis* from mining area (MA) (**a**–**c**) and non-mining area (nMA) (**d**,**e**) in Yunnan Province, southwestern China. Microsclerotia (M) (**a**,**d**) and hyphae (H) (Figure 1**e**) of DSE structures and vesicles (V) (**c**,**d**), arbuscules (A) (**b**) and hyphal coils (HC) (**b**) of AMF structures are visible in the micrograph.

**Figure 2 plants-10-02731-f002:**
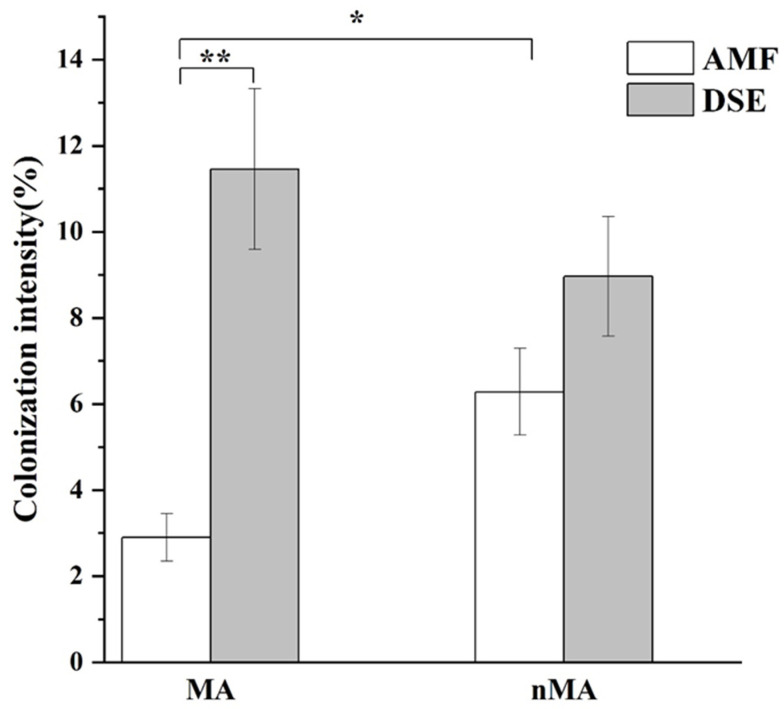
The total colonization intensity of DSE and AMF in the roots of *O. sinensis* from MA and nMA areas in Yunnan Province, southwestern China (means ± SE, *n* ≥ 10). * Indicates significant differences between two groups (*t*-test, * *p* < 0.05, ** *p* < 0.01).

**Figure 3 plants-10-02731-f003:**
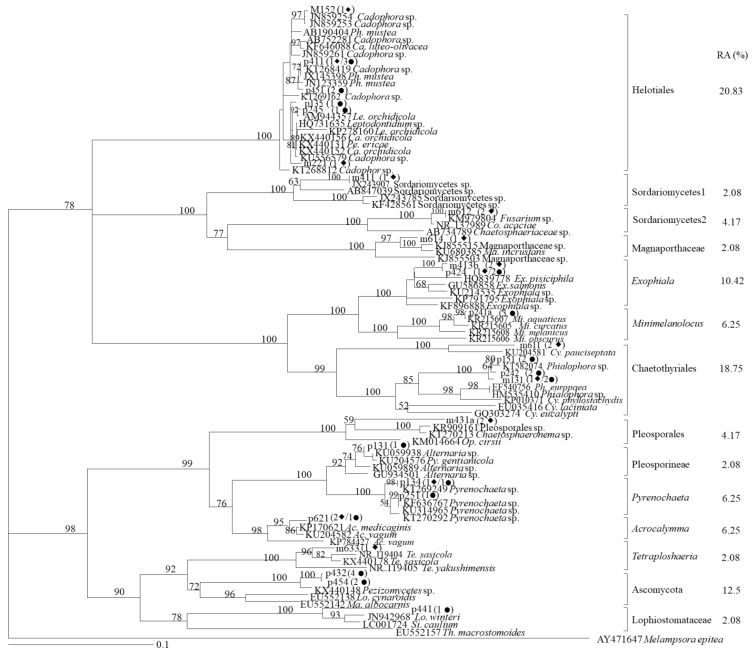
Molecular phylogeny of ITS sequences of melanized fungal isolates from the roots of *O.*
*sinensis* grown in Huize MA and nMA areas in Yunnan Province, southwestern China. ◆ and ● in parentheses mean the numbers of fungal strains isolated from the MA and nMA, respectively, and the relative abundance (RA) of each clade was calculated. Numbers above the nodes denote 1000 bootstrap support in neighbor-joining analysis.

**Figure 4 plants-10-02731-f004:**
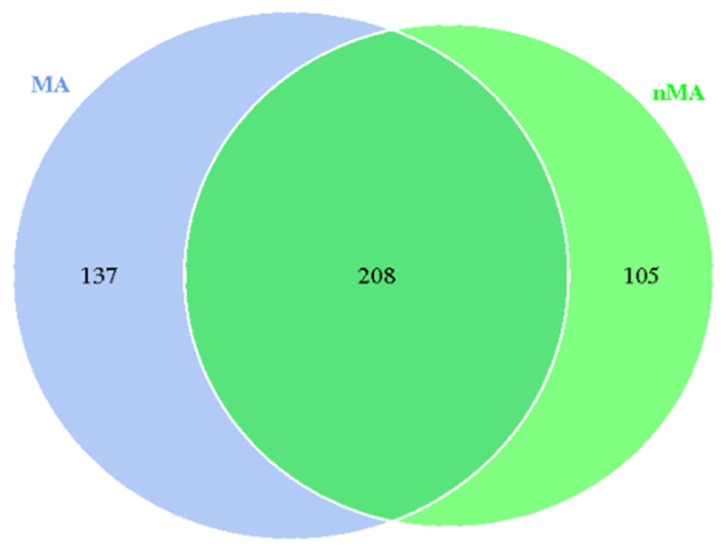
Venn diagram depicting the distribution of endophytic fungi colonizing in the roots of *O. sinensis* in MA and nMA areas in Yunnan Province, southwestern China (OTU level).

**Figure 5 plants-10-02731-f005:**
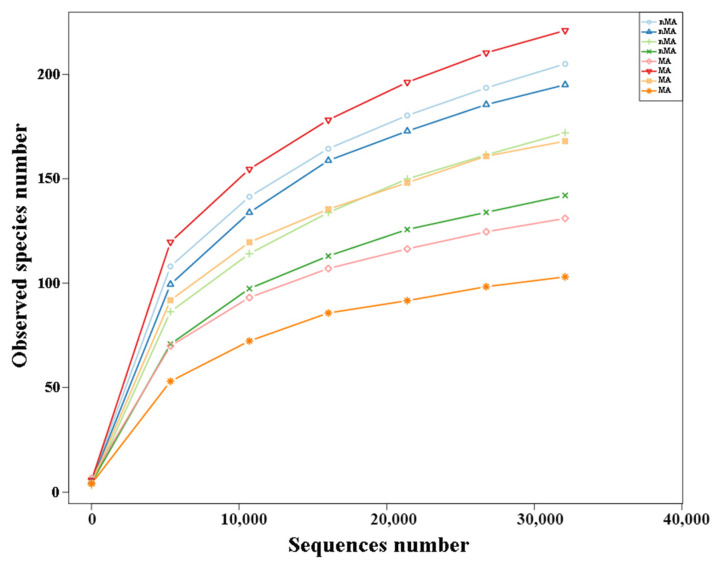
Rarefaction curve of endophytic fungi colonizing *O. sinensis* roots from MA and nMA areas in Yunnan Province, southwestern China.

**Figure 6 plants-10-02731-f006:**
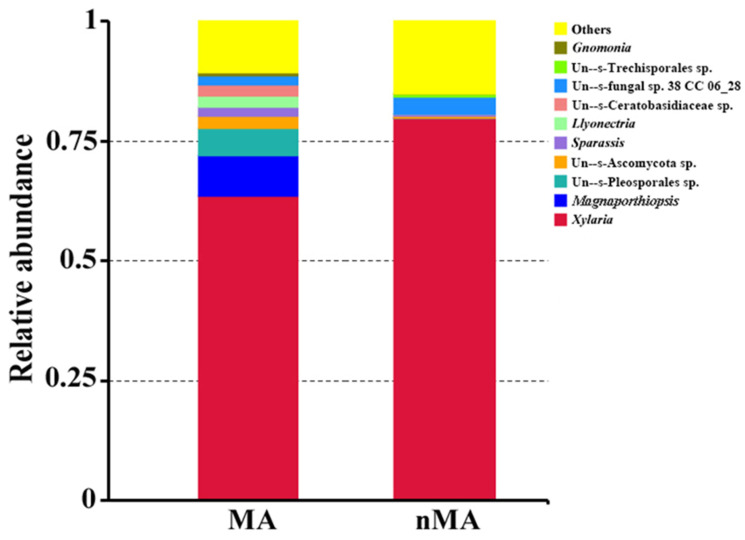
Community composition and relative abundance of endophytic fungi colonizing *O. sinensis* roots of MA and nMA areas in Yunnan Province, southwestern China (genus level).

**Figure 7 plants-10-02731-f007:**
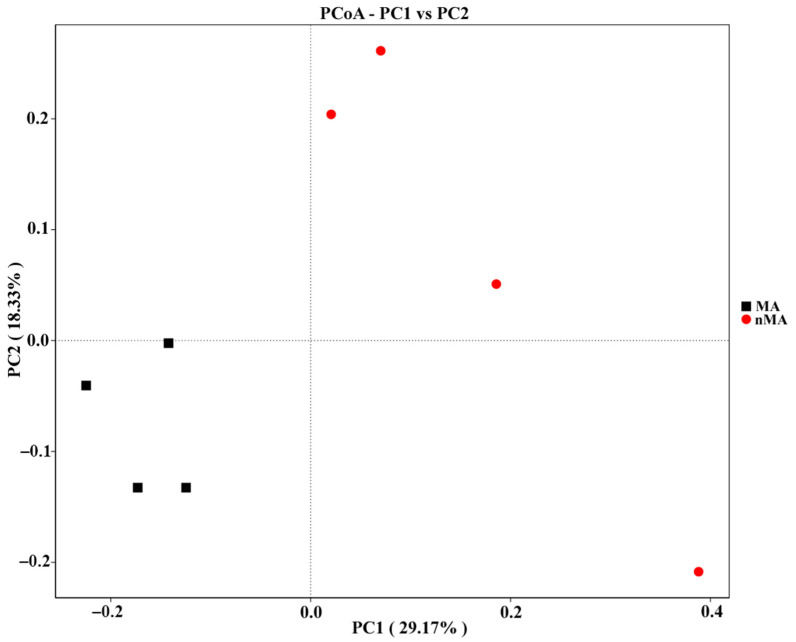
Principal Coordinate Analysis (PCoA) of endophytic fungal community structures in the roots of *O. sinensis* in the MA and nMA areas in Yunnan Province, southwestern China. Each red symbol represents an nMA sample, and each black symbol represents an MA sample.

**Figure 8 plants-10-02731-f008:**
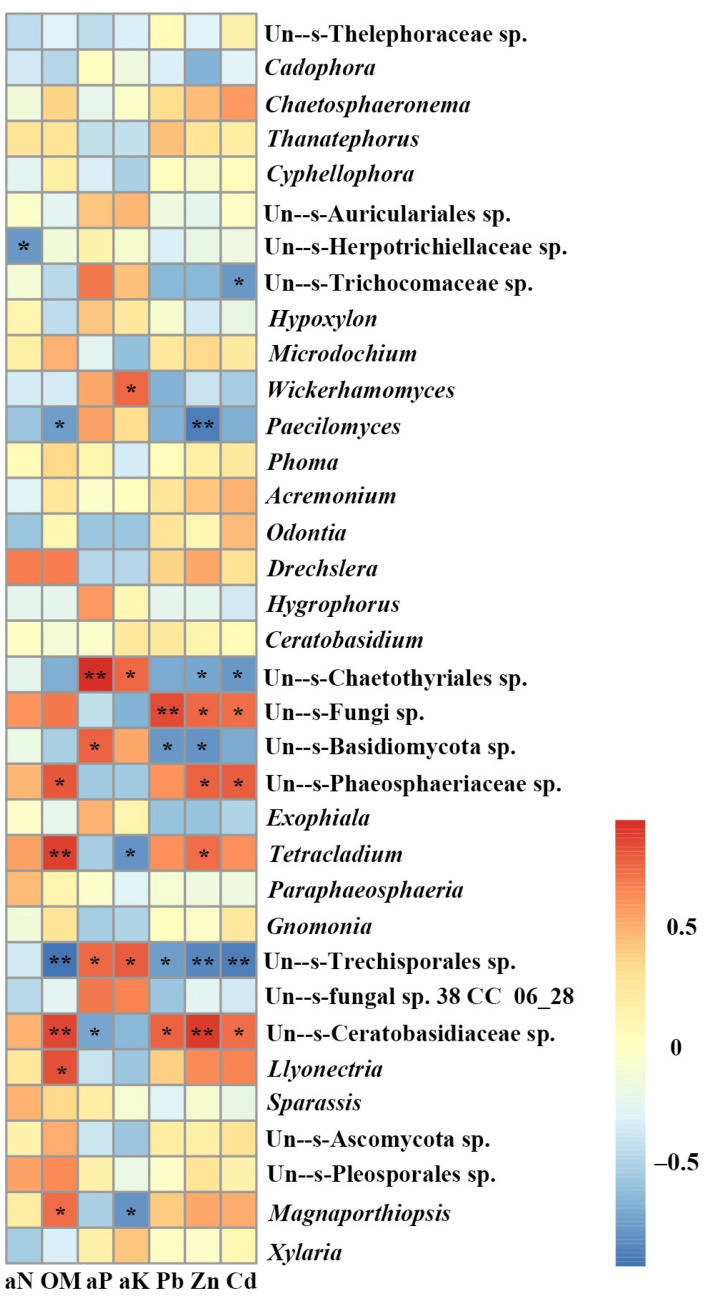
Spearman’s correlation analysis between environmental factors and endophytic fungal community structures in the roots of *O. sinensis* in the MA and nMA areas in Yunnan Province, southwestern China (Genus level). aN, alkaline hydrolysis nitrogen; OM, organic matter; aP, available phosphorus; aK, available potassium. * Indicates a significant correlation between environment factor and endophytic fungi (Spearman correlation analysis, * *p* < 0.05, ** *p* < 0.01).

**Table 1 plants-10-02731-t001:** Comparison of the physiochemical properties and lead (Pb), zinc (Zn) and cadmium (Cd) concentrations in the rhizosphere soil samples from both mining (MA) and non-mining (nMA) areas in Yunnan Province, southwestern China (means ± SE, *n* ≥ 3).

Sites	Physiochemical Properties (mg kg^−1^)					Total Metal Concentration (mg kg^−1^)		
	Alkaline Hydrolysis N	Organic Matter	Available P	Available K	pH	Pb	Zn	Cd
MA	60.4 ± 13.10	16.9 ± 3.47 b	19.7 ± 0.84 a	184.9 ± 25.03	6.2 ± 0.06	191.9 ± 37.67 b	719.5 ± 210.00 b	1.9 ± 0.51 b
nMA	42.5 ± 2.80	5.3 ± 0.72 a	26.1 ± 1.06 b	232.0 ± 8.18	6.2 ± 0.03	31.9 ± 3.87 a	192.5 ± 25.40 a	0.0 ± 0.00 a

Different letters mean significant differences according to *t*-test (*p* < 0.05).

**Table 2 plants-10-02731-t002:** Diversity indices of endophytic fungi colonizing in the roots of *O. sinensis* from MA and nMA areas in Yunnan Province, southwestern China (*n* = 4).

Sites	ACE	Chao1	Shannon	Simpson
MA	235.1 ± 18.23	231.7 ± 16.21	2.7 ± 0.31	0.7 ± 0.08
nMA	196.3 ± 32.88	191.4 ± 31.70	3.0 ± 0.22	0.8 ± 0.04

## Data Availability

All data generated or analyzed during this study are included in this published article.

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
