# Peer review of "Biodiversity of Root Endophytic Fungi from Oxyria sinensis Grown in Metal-Polluted and Unpolluted Soils in Yunnan Province, Southwestern China"

_plants, 2021, doi:10.3390/plants10122731_

Round 1

Reviewer 1 Report

Reviewer #2

Comment 1: Line 4: Is there an author name missing? If not remove “and”

Response: Thank you, we made necessary change.

Comment: Noted.

Comment 2: Line 58-65: These 2 sentences are redundant and convey the same meaning. You may change it to make meaning flow.

Response: Thank you for the reviewer’s suggestion. We have rephrased the sentence in line no.

56-59.

Comment: Cleared and noted.

Comment 3: Line 58-71: This section can be misleading to the readers as you write extensively

about phytoremediation of heavy metals and the root-associated endophytes playing a role in it.

However, in the study there is no temporal result showing that phytoremediation does occurs in

  1. sinensis.

Response: Many thanks for the constructive comment. We have rephrased this paragraph, and

deleted the extensive topics on the phytoremediation of heavy metals. Please see lines 56-61.

Comment: Noted the changes.

Comment 4: Line 63-65: Grammar is inconsistent. Change the sentence structure.

Response: The sentence has been reconstructed according to the reviewer’s suggestions. Please

see lines 56-61.

Comment: Noted the changes

Comment 5: Line 134-135: Figure 3: the phylogenetic tree looks very cluttered. The strains can

be in the same font as the closely related members albeit darker (bolder).

Response: Many thanks for your constructive comment. We re-analyzed the data, and the fungal

origins and their relative abundance have been supplemented in figure 3 in line 130 on page 6.

Comment: Noted.

Comment 6: Line 182-185: Explain the meaning of the stars in the figures? Are they above a

certain value correlation value?

Response: Necessary changes have been made. Please see lines 183-184. Thanks.

Comment: Noted.

Comment 7: Line 188-190: Redundant. Remove sentence.

Response: We removed the sentence. Please see lines 186-187. Thank you.

Comment: Noted.

Comment 8: Line 217-220: “….. and fungal species such as Helotiales,….” Heloitales is an order

name and not a species name. “ Exophiala” is a genus. This sentence can be restructured to

something of the like: “Fungal species belonging to the Helotiales order ……”

Response: As per the reviewer’s suggestion, we have re-phrased the sentence please see lines

214-217. Many thanks!

Comment: Noted.

Comment 9: Line 221-238: There seems to be a section missing in the destination section of the

destination of these heavy metals. Do they keep persisting in the environment? Do they get

bioaccumulated in the plants or the AMF or DSE?

Response: Thank you for the valuable suggestions. We have rephrased this topic discussion.

Please see lines 220-232.

Comment: Noted.

Comment 10: Line 261-263: References missing for the methods used.

Response: The references now have been updated in the text and the reference list. Please see

lines 268-273. Thanks.

Comment: Noted.

Comment 11: Line 294-297: It is not clear as to how the root was treated prior to total DNA

extraction? It is of utmost importance to include the pretreatment of the roots prior to DNA

extraction. The extrapolation of your findings are based on whether you are including or

excluding ectomycorrhizal fungi. The claim on Figure 6 is only on endophytic fungi colonization

– include how the root was treated to remove ectomyccorhizal fungi.

Response: The detailed information on the pre-treatments of the roots, e.g., thoroughly cleaned

and surface-sterilization, prior to total DNA extraction was added. Please see lines 303-308.

According to the literatures, no ectomycorrhizal fungi were reported in the roots of O. sinensis.

Thus, no mycorrhizal anatomical examination was conducted. In the present study, the fungi

belonging to Ceratobasidiaceae family were also found in the roots of O. sinensis, which was

identified as pathogens, saprotrophs, non-mycorrhizal endophytes, as well as orchid mycorrhizal

and ectomycorrhizal symbionts (for example, please see reference: Veldre V, Abarenkov K,

Bahram M, Martos F, Selosse MA, Tamm H, Koljalg U, Tedersoo L. Evolution of nutritional

modes of Ceratobasidiaceae (Cantharellales, Basidiomycota) as revealed from publicly available

ITS sequences. Fungal Ecology 6 (2013) 256–268). So, keeping in mind that the anatomic

characteristics must be assessed prior to the further treatments.

Comment: Noted. Thanks for clarifying this important point.  

Comment 12: Line 329-330: Sentence grammar is off. Change.

Response: As per the reviewer’s suggestion, we have re-phrased the sentence, please see lines

340-348. Many thanks!

Comment: Noted.

Supplementary Materials:

Comment 13: The main body of the paper did not reference the supplementary section. Why

include it? if you want to add it reference the paper.

Response: No supplementary materials were provided in our manuscript. We have removed it.

Thanks.

Comment: Okay.

Total colonization intensity Tab:

Comment 14: Colonization intensity of DSE under nMA- values are missing for last 2 samples.

Are these are “0” or “NA”? Explain these missing values

Response: In the present study, ten sample replications were collected and the fungal

colonization intensity was assessed. Prior to statistical analyses, the normality and homogeneity

of variance were assessed, and two outliers of DSE colonization intensity under nMA- values

were detected. According to the reference by Kutner et al. (2004), two outliers were screened

and deleted, and excluded them from the dataset in each comparison. Please see line no. 334-

  1. Thanks.

Reference: Kutner, H. M., Nachtsheim, J.C., Neter, J., Li, W. Diagnostics and Remedial

Measures Applied Linear Statistical Models (5th ed.), McGraw-Hill/Irwin: New York, America,

2004, pp. 108, ISBN 0-07-238688-6

Comment: Noted.

Comment 15: Colonization intensity of AMF under nMA – one value missing. Explain.

Response: In the present study, ten sample replications were collected and the fungal

colonization intensity was assessed. Prior to statistical analyses, the normality and homogeneity

of variance were assessed, and one outlier of AMF colonization intensity under nMA- value was

detected. According to the reference by Kutner et al. (2004), the outlier was screened and deleted,

and excluded them from the dataset in each comparison. Please see line no. 334-336. Thanks.

Reference: Kutner, H. M., Nachtsheim, J.C., Neter, J.,Li, W. Diagnostics and Remedial

Measures Applied Linear Statistical Models (5th ed.), McGraw-Hill/Irwin: New York, America,

2004, pp. 108, ISBN 0-07-238688-6.

Comment: Noted.

Reviewer 2 Report

Comments in the last review has got appropriate deal in the revised manuscript (plants-1493000) and the Response to the reviewers-3’ comments from authors. Thank you for your consideration. Also, reviewer would like to point out trivial comments of typing errors.

・line 102,  "non-mining" would be "mining".

・line 107-108,  "figs. d-c" would be "figs. d-e".

This manuscript is a resubmission of an earlier submission. The following is a list of the peer review reports and author responses from that submission.

Round 1

Reviewer 1 Report

The paper by Meiyan Zhu et al. describes the fungal diversity of root endophytes from O. sinensis in metal polluted soils.

Both arbuscular mycorrhizal fungi (AMF) and dark septate endophytes (DSE) are described/identified and show different accumulation in roots as a function of the metal pollution.  

This study is mainly descriptive and just correlates the presence of many fungi with the soil type (polluted/non polluted).

The paper is not well written. Figure legends are incomplete and do not allow understanding the figures. See for example Figure 1 in which there are 5 panels (a-e) but we do not know to what they correspond.  Are they root from contaminated or not soils?

The comments about Table1 do not correspond to what is observed in this table. The authors say higher available P in contaminated soils, but this is the opposite in the Table. They also say that there is no difference for organic matter between the two soil, when the values are 16,9 versus 5,3.

There are also problems in the description of the results, the authors giving the different values at the end of the paragraphs after that they have described the differences between the samples.

In section 2.3 they describe the relative abundance of the fungi but refer to a phylogenetic tree. This is not the type of figure describing abundance.

In summary I believe that this paper is not properly written and cannot be published as it is.

Reviewer 2 Report

Comments:

Line 4: Is there an author name missing? If not remove “and”

Line 58-65 : These 2 sentences are redundant and convey the same meaning. You may change it to make meaning flow.

Line 58:71: This section can be misleading to the readers as you write extensively about phytoremediation of heavy metals and the root-associated endophytes playing a role in it. However, in the study there is no temporal result showing that phytoremediation does occurs in O sinensis

Line 63-65: Grammar is inconsistent. Change the sentence structure.

Line 134-135: Figure 3: the phylogentic tree looks very cluttered. The strains can be in the same font as the closely related members albeit darker (bolder).

Line 182-185: Explain the meaning of the stars in the figures? Are they above a certain value correlation value?

Line 188-190: Redundant. Remove sentence

Line 217-220: “….. and fungal species such as Helotiales,….” Heloitales is an order name and not a species name. “ Exophiala” is a genus. This sentence can be restructured to something of the like: “Fungal species belonging to the Helotiales order ……”

Line 221-238: There seems to be a section missing in the destination section of the destination of these heavy metals. Do they keep persisting in the environment? Do they get bioaccumulated in the plants or the AMF or DSE ?

Line 261-263: References missing for the methods used.

Line 294-297: It is not clear as to how the root was treated prior to total DNA extraction? It is of utmost importance to include the pretreatment of the roots prior to DNA extraction. The extrapolation of your findings are based on whether you are including or excluding ectomycorrhizal fungi. The claim on Figure 6 is only on endophytic fungi colonization – include how the root was treated to remove ectomyccorhizal fungi.

Line 329-330: Sentence grammar is off. Change.

Supplementary Materials:

The main body of the paper did not reference the supplementary section. Why include it ? if you want to add it reference the paper.

Total colonization intensity Tab:

  • Colonization intensity of DSE under nMA- values are missing for last 2 samples. Are these are “0” or “NA”? Explain these missing values
  • Colonization intensity of AMF under nMA – one value missing. Explain.

Reviewer 3 Report

Reviewer would like to point out following comments.

・Figure 1,  It should be clearly shown which photos (a-e) shows root sample from MA or nMA.

・Figure 3,  Molecular phylogeny is arranged by combination of several taxonomic ranks as shown in right side of this figure. Why didn’t authors unify them?

・line 232-234 and 329-330,  Authors compared the endophytic fungi community structure in the roots of Oxyria sinensis collected from a mining area (MA) and a neighboring non-mining area (nMA), discussing the function of the fungi community directly connected with survivability of this plant in the mining condition. However, experimental data did not directly indicate the function of detected fungi to the plants. Effects of bacterial community or other environmental conditions (ex., essential elements except N, P, K) are not mentioned here at all. Basic experiments combining detected fungi or their consortia with O. sinensis would be contributory to the solution of this question.